# A Multiparametric MRI-Based Radiomics Analysis to Efficiently Classify Tumor Subregions of Glioblastoma: A Pilot Study in Machine Learning

**DOI:** 10.3390/jcm10092030

**Published:** 2021-05-10

**Authors:** Fang-Ying Chiu, Nguyen Quoc Khanh Le, Cheng-Yu Chen

**Affiliations:** 1Research Center for Sustainable Development Goals (SDGs), Tzu Chi University, Hualien 970374, Taiwan; fychiu@gms.tcu.edu.tw; 2Professional Master Program in Artificial Intelligence in Medicine, College of Medicine, Taipei Medical University, Taipei 106339, Taiwan; 3Research Center for Artificial Intelligence in Medicine, Taipei Medical University, Taipei 106339, Taiwan; 4Translational Imaging Research Center, Taipei Medical University Hospital, Taipei 110301, Taiwan; 5Department of Radiology, School of Medicine, College of Medicine, Taipei Medical University, Taipei 110301, Taiwan; 6Department of Medical Imaging, Taipei Medical University Hospital, Taipei 110301, Taiwan

**Keywords:** glioblastoma, MRI, quantitative imaging, oncologic imaging, radiomics, texture analysis, ground truth, machine learning, precision medicine

## Abstract

Glioblastoma multiforme (GBM) carries a poor prognosis and usually presents with heterogenous regions of a necrotic core, solid part, peritumoral tissue, and peritumoral edema. Accurate demarcation on magnetic resonance imaging (MRI) between the active tumor region and perifocal edematous extension is essential for planning stereotactic biopsy, GBM resection, and radiotherapy. We established a set of radiomics features to efficiently classify patients with GBM and retrieved cerebral multiparametric MRI, including contrast-enhanced T1-weighted (T1-CE), T2-weighted, T2-weighted fluid-attenuated inversion recovery, and apparent diffusion coefficient images from local patients with GBM. A total of 1316 features on the raw MR images were selected for each annotated area. A leave-one-out cross-validation was performed on the whole dataset, the different machine learning and deep learning techniques tested; random forest achieved the best performance (average accuracy: 93.6% necrosis, 90.4% solid part, 95.8% peritumoral tissue, and 90.4% peritumoral edema). The features from the enhancing tumor and the tumor shape elongation of peritumoral edema region for high-risk groups from T1-CE. The multiparametric MRI-based radiomics model showed the efficient classification of tumor subregions of GBM and suggests that prognostic radiomic features from a routine MRI exam may also be significantly associated with key biological processes that affect the response to chemotherapy in GBM.

## 1. Introduction

Glioblastoma multiforme (GBM) is characterized by its rapid growth and infiltration into the surrounding brain parenchyma. In adults, it occurs most often in the cerebral hemispheres, especially in the frontal and temporal lobes. The survival rate of brain tumor patients is only approximately 14.2 months after diagnosis [1]. The 2016 World Health Organization Classification of Tumors of the Central Nervous System (CNS WHO) presents a major restructuring of the diffuse gliomas and incorporates new entities defined using both histological and molecular features, including glioblastoma, isocitrate dehydrogenase (IDH)-mutant status, and 1p/19q codeletion [2].

Several imaging modalities can help diagnose and locate the recurrence of brain tumors, such as computed tomography (CT), magnetic resonance imaging (MRI), and positron emission tomography (PET) scans. Dynamic contrast-enhanced (DCE) CT/MR perfusion provides functional information about the neoangiogenesis of the tumor. Diffusion MR imaging can enunciate tumor cellularity and the integrity of white matter. MR spectroscopy (MRS) is used to study the chemical profile of tumors and oncogene markers such as IDH mutation. Of these, conventional MRI images providing multi-parametric tissue contrast are the most commonly used modality to assess the treatment response compared with other imaging modalities. Brain MRI not only helps describe the tumor location and morphology for semiautomated contouring but also offers ground truth data for machine learning.

Radiomics is an emerging technique for the characterization of tumor phenotypes through extracting and quantifying thousands of imaging features such as geometrical and morphological characteristics, cluster shade, intensity, diffusion kurtosis, and texture or wavelet ascribed to the cancer phenotype [3]. This technique is becoming increasingly popular in data mining, especially with the urgent need for medical imaging analysis. Briefly, an optimal radiomics analysis involves four steps: (1) imaging acquisition, (2) mark up or segmentation, (3) feature extraction, and (4) statistical analysis. The resulting features can be used to inform imaging diagnosis, predictive, and prognostic factors for therapy response in oncology [4,5,6]. In addition, combining anatomical and functional imaging to analyze radiomics features can help in the classification of imaging biomarkers in oncology. The peculiarity of GBM is its heterogeneous regions containing different cell types (including blood vessels) and areas of dead cells (necrosis) that are locally invasive, and these characteristics can be used for feature extraction.

Recently, machine learning and deep learning models have been developed to analyze the radiomic features of, and classify, GBM regions. Such computational models in biomedical imaging aim to learn information directly from data and can have various applications, such as sub-typing in diagnosis and determining tumor margins for surgical planning. In 2017, segmented labels and radiomics features of high- and low-grade glioma datasets of The Cancer Genome Atlas (TCGA) (i.e., TCGA-GBM and TCGA-LGG) were released [7], which attracted considerable attention to this segmentation task, and numerous models have been proposed, such as convolutional neural networks [8] and U-net architecture [9]. To obtain the radiomic features affecting GBM segmentation, Lu et al. [10] proposed a support vector machine framework for the three-layer classification of GBMs. Wu et al. [11] developed a random forest-based approach for IDH genotype prediction using radiomic features. Different analyses have been performed to establish optimal radiomics features for GBM imaging features to predict molecular subtypes and the prognosis of tumor [12,13,14].

In this study, we developed a radiomics-based approach to explore which radiomic features and machine learning algorithms are the most helpful in the tissue classification of patients with GBM. Radiomic features were computed on four different GBM tissue regions—necrosis, solid part, peritumoral tissue, and peritumoral edema. Classification and feature selection were performed using different machine learning and deep learning techniques. We established important radiomic features, which provide a better understanding of GBM classification and novel clinical insights into GBM characteristics for personalized precision medicine.

## 2. Materials and Methods

A multiparametric MRI-based radiomic analysis (Figure 1). The resulting features can be used to inform imaging diagnosis, predictive, and prognostic evaluation for treatment selection in precision medicine.

### 2.1. Patients

This study recruited 23 patients with GBM from local hospitals after receiving institutional review board approval (TMU-JIRB No. 201603086). All patients suspected of having cerebral GBM based on conventional radiologic findings were enrolled in this prospective study before any treatment (10 women, 13 men; age range, 42–83 years; average age, 62.60 years), and imaging was performed between October 2014 and February 2019. The inclusion criteria were as follows: (1) neuropathological evaluation following surgery or stereotactic biopsy, with all lesions being histopathologically confirmed grades IV glioma, classified as per the 2016 CNS WHO [2], and (2) available preoperative MRI consisting of gadolinium-based contrast-enhanced T1-weighted images (T1-CE), T2-weighted images (T2-WI), T2-weight fluid-attenuated inversion recovery (T2-FLAIR), and apparent diffusion coefficient (ADC) images. Moreover, we also retrieved 31 patients with GBM from The Cancer Imaging Archive (TCIA) database as our validation cohort (Table 1). This dataset was released by BraTS challenge, in which the MRIs were also used to segment GBM patients [15].

### 2.2. Data Processing and Region of Interest (ROI) Segmentation

All patients underwent routine MRI scans. The segmentation is performed slice-wise, where the input data include the T1-CE, T2-WI, T2-FLAIR, and ADC MR images of each patient. All methods were applied to the T1 postcontrast images using default parameters, except for machine learning models that have no tunable parameters. The manual adjustment was performed, if demanded, by an experienced researcher in neuroradiology (F-Y C). Additionally, the experienced neuroradiologist recognized after segmentation MR Imaging (C-Y C).

### 2.3. Extraction of Radiomics Features

This study implemented semiautomatic annotation for radiomic features using a previously described programming system [10] in MATLAB. Radiomic features from slice-by-slice segmentations were extracted by an experienced neuroradiologist. To ensure the reliability of quantitative imaging features, tumor contouring by manual delineation to separate four different regions of tissues, twice for each case, was performed. The manually corrected segmentation features were extracted from four subregions: necrosis, solid part, peritumoral tissue, and peritumoral edema. The feature dataset was divided into three groups: (1) histogram: the first-order statistics computed from pixels and voxel intensities, (2) geometry: the three-dimensional morphological characteristics of the tumor, and (3) texture: second- and high-order spatial distributions of the intensities in the image. The texture features were extracted using several methods, including features based on the gray-level co-occurrence matrix (GLCM) and gray-level run-length matrix (GLRLM). GLCM is the most commonly used texture feature, which considers only voxels within a specific range of gray value; it can produce a matrix of the spatial relationships [4,16] and considers the arrangements of pairs of voxels to calculate texture indices. GLRLM is a matrix of all the voxels within the same gray-level value [17], which gives the size of homogeneous runs for each one. Determining texture matrix representations requires the voxel intensity values within the volume of interest to be discretized. This step reduces image noise and normalizes intensities across all images [18]. We adopted local binary pattern (LBP) histogram for texture; this summarizes the texture characteristics and patterns of a surface into a histogram, which can be used as input for pattern classification.

Radiomic features were extracted as described previously [10]. A total of 1316 features on the raw MR images were selected for each annotated area, including 128 intensity histogram (64 first-order and 64 histogram features of LBP), 32 geometry features (shape- and size-based features), 132 texture features (88 GLCM features and 44 GLRLM features), and 1024 scale-invariant feature transform features. All the features were combined and used as the input radiomic features for the machine learning model.

### 2.4. Machine Learning Algorithms and Measurement Metrics

To evaluate the performance of our model, we compared different machine learning (such as k-nearest neighbors (kNN), naïve Bayes (NB), and random forest) and deep learning (deep neural network (DNN)) techniques. The hyperparameters of all classifiers were also optimized using a grid search with cross-validation (gridsearchCV) to reach the best performance, to ensure a fair comparison. As a detail, we ran the kNN algorithm with a different number of neighbors (k = 1…30) and three distance learning metrics (euclidean, manhattan, minkowski). For random forest, we used the number of features ranging from 100 to 2000 (stepsize = 100), and maximum depth values ranging from zero to 110 (stepsize = 10). Finally, the DNN was tuned with a different number of layers and parameters. Furthermore, different feature selection techniques were examined to reduce the noise of features and the computational complexity of the machine learning model. All the models were implemented using the Python programming language with scikit-learn and keras packages.

Due to the limited data, leave-one-out cross-validation was used to validate the overall performance [19]. In this approach, each sample is used as a test set while the others are the training set for that sample, and the reported accuracy is the mean of all testing accuracy values. The accuracy of classification was evaluated with majority vote (i.e., a threshold cutoff of 50%). We also adopted different performance metrics in the prediction model, such as recall, precision, F1-score, accuracy, and area under the curve (AUC) to stratify the training data to improve machine learning-based brain tumor region classification.
(1)$$\mathrm{Recall}=\frac{TP}{TP+FN}$$
(2)$$\mathrm{Precision}=\frac{TP}{TP+FP}$$
(3)$$F1=2\cdot \frac{\mathrm{Precision}\cdot \mathrm{recall}}{\mathrm{Precision}+\mathrm{recall}}$$
(4)$$\mathrm{Accuracy}=\frac{TP+TN}{TP+FP+TN+FN}$$

Here, *TP* is the number of true positives; *TN* is the number of true negatives; *FP* is the number of false positives; and *FN* is the number of false negatives. Moreover, two testing methods (Spearson and Spearman) were applied to show the significance of our radiomics features. Statistical analysis and machine learning were implemented using different packages with the Python programming language.

## 3. Results

### 3.1. Ground Truth Segmentation and Identification of Tumor Habitats on MRI

(Figure 2 and Figure 3) display tumor habitats which are color-coded and overlaid on T2-FLAIR and T1 and T2 annotated images for the ground truth and semantic features from GBM. First, different GBM regions were accurately labeled into four ROIs and joint intensity color-maps on T2-FLAIR: necrosis (red), solid part (orange), peritumoral tissue (yellow), and peritumoral edema (green). Second, computational features were extracted using annotated imaging; both color-coded semantic and features were derived from multimodal MR radiomics to analyze GBM texture features to compare tumor characteristics such as homogeneity and entropy, and correlate them with spatial-habitat imaging (Figure 4). All features were combined and fed into the machine learning algorithms.

### 3.2. Classification of GBMs Using Different Machine Learning and Deep Learning Algorithms

We evaluated the performance of our GBM classification using kNN, Naïve Bayes, random forest, and DNN models. The input was a set of 329 radiomic features (as mentioned in Section 2.2). For a fair comparison, we used the grid search strategy (with leave-one-out cross-validation) to determine the optimal parameters for the models. The average accuracy of kNN, Naïve Bayes, random forest, and DNN classifiers was 55.9%, 80.6%, 91.4%, and 67.7%, respectively, indicating that the random forest model outperformed others.

Because our models were trained in multiclassification, we presented the performance results using ROC curves and AUC (Figure 5). For the random forest model, all GBM regions demonstrated similar performance curves with extremely high sensitivity and specificity (Figure 5c). The other classifiers had inconsistent performance, which means that they sometimes could not classify some regions correctly, especially the solid part (accuracy: 90.4%) and edematous region (accuracy: 90.4%) (Table 2).

Two testing methods (Pearson and Spearman) were applied, and the testing results are shown in (Figure 6). According to this list, some radiomics features were consistent and highly correlated to the ROI labels, such as LBP_Uniformity, surface-to-volume ratio, spherical disproportion, and LBP_Skewness. These features might play essential roles in deciding on different regions of GBMs. This list also revealed some overlapping information between our features and previous works on radiomics-based classification.

To clarify the metrics related to the random forest model for each region, we dissected our multiclassification into four binary classifications. As presented (Table 2), all of the measurement metrics for four regions reached significant levels and compare with the small subset of TCIA dataset; thus, we considered our features to be helpful in efficient GBM classification from different modalities and data sources. Furthermore, the overall performance was consistent with multiclassification performance.

### 3.3. Selection Approach for Radiomics Features Using Random Forest

The most important concern in machine-learning-based classifiers using radiomics features is the dimensions of data. The large amount of information used as feature sets for the models will increase the model complexity and may lead to overfitting. Therefore, reducing the number of features is essential, which can be achieved by commonly used methods, such as principal component analysis and clustering (k-means or hierarchical). However, this study tested the possibility of supervised learning in identifying the optimal features—the top-ranking radiomic features that might affect the classification model. Thus, we attempted to determine the important features when training the random forest algorithm. We arranged the important scores (*IS*) from the highest to lowest: the top-rank scores were from the following features: LBP_Mean (*IS* = 0.058151), LBP_Entropy (*IS* = 0.040182), and LBP_Kurtosis (*IS* = 0.037939). Twenty top-rank features generated from random forest were used as inputs to evaluate the correlations of these features. The models were trained for classification among four tumor subregions, consisting of necrosis, solid part, peritumoral, and edema as accuracy percentages of the corresponding values in 93.6%, 90.4%, 95.8%, and 90.4% for each target outcome, respectively.

### 3.4. Statistical Analysis

Statistical analysis and machine learning were implemented using different packages with the Python programming language. Two testing methods (Pearson and Spearman) were applied, and the testing results are shown in (Figure 6). According to this list, some radiomic features were consistent and highly correlated to the ROI labels, such as LBP_Uniformity, surface-to-volume ratio, spherical disproportion, and LBP_Skewness. These features might play essential roles in deciding different regions of GBMs. This list also revealed some overlapping information between our features and previous works on radiomic-based classification.

## 4. Discussion

Medical imaging is becoming the cornerstone for managing and evaluating the treatment response to cancer. Radiomic-based approaches provide new insights into disease characteristics, particularly in the field of oncology. However, well-validated noninvasive biomarkers that can reflect underlying tumor habitats to provide information to guide therapy are lacking. GBM is primarily diagnosed by neuroimaging, followed by biopsy or resection of the tumor tissue to diagnose, characterize, and grade and stage the tumor. Confirmatory analysis can be achieved using immunohistochemistry and molecular analyses, including 1p/19q codeletion, IDH1 mutation, and/or expression of p53 and O^6^-methylguanine methyltransferase hypermethylation in epigenetic alterations [20,21]. The present study leveraged radiomic features based on imaging signatures of the heterogenous GBM tumor tissue parts and created a radiomic-based model for the semiautomatic annotation of GBM using MRI, ground truth, and machine learning.

Our study classified GBMs into four regions—necrosis, solid part, peritumoral tissue, and peritumoral edema. This criterion was different from most studies on the segmentation of GBM since they all defined three regions: necrosis, solid part, and peritumoral edema [11,22,23]. Notably, the fourth region—peritumoral tissue—performed well in our model, with an accuracy reaching 95.8%, which was 2%–5% better than those of the other three regions (Table 2). This implies that this region was useful for extracting more radiomic features for the segmentation and diagnosis of GBM.

Traditionally, machine learning methods in the radiomics domain have usually been separate from the feature selection techniques [11,16]. However, we used supervised learning to determine the optimal set of radiomic features. Notably, the important radiomics features were obtained from the original model, not from the other unsupervised learning techniques. Thus, our radiomic features are reliable and more suitable to our selected model. We also validated different machine-learning and deep-learning techniques to establish the most useful model for generating the optimal radiomics feature set. Our radiomic features (LBP_Uniformity, surface-to-volume ratio, spherical disproportion, and LBP_Skewness) differ from previous works on GBM using traditional feature selection techniques [18,23,24] and may guide further research on the segmentation of GBMs.

Semiautomatic segmentation currently represents the classification for response assessment using volumetric measurements that may capture tumor geometry more accurately; this is particularly useful for GBMs, which are often irregularly shaped. Moreover, large-scale studies have established the benefits of using volumetry-based radiomics features for tumor segmentation compared with complicated verification approaches [25].

Semiautomatic segmentation has three crucial challenges for accurate features. The first challenge is variability in tumoral extraction: an image preprocessing analysis that separates the tumor from normal brain tissue is essential for many neuroimaging applications. This needs to be compared with different neuroimaging biomarkers for quantitative and tumoral heterogeneity surrogates as a reference. DCE MR perfusion provides functional information regarding the tumor hemodynamic status such as relative cerebral blood volume (rCBV), and transfer constant (K^trans^) was evaluated as a quantitative biomarker; it also can help in the differential diagnosis by tumor recurrence and radiation necrosis [26,27]. Diffusion-weighted MR imaging together with ADC maps reflect hindered and restricted diffusion pools within a voxel on diffusion-weighted MR images. This is always in the routine diagnostic protocol, but diffusion kurtosis imaging is usually reserved for research purposes [28]. Within a specific ROI, areas of necrosis and peritumoral edema can increase ADC values, which increase cellular density architecture due to the reduced ADC values [27]. Hilario et al. [29] reported that combined maximum rCBV and minimum ADC values improve the accuracy of preoperative MRI grading of gliomas. The ADC threshold value of 1185 × 10^−6^ mm^2^/s has a sensitivity of 97.6% and specificity of 53.1% in the discrimination of high-grade (grades III and IV) and low-grade (grade II) gliomas. Regarding tumor physical processes, MRS can be used to study tumor biomarkers such as IDH mutations, morphological image processing for quantitative analysis, and metabolism in noninvasive investigations of the link between the molecular basis of GBM and imaging attributes [27,28]. Most high-grade tumors have greater choline concentrations in astrocytoma. By contrast, the typical spectroscopic characteristics of lower-grade tumors and normal brain tissue include high choline, lactate, and lipid concentrations and low *N*-acetyl aspartate and myoinositol concentrations [30].

The second challenge is generalizability: MR intensity values vary substantially depending on the MR scanner properties and acquisition parameters, such as scanner type protocol and contrast agent injection rate, respectively, resulting in substantial differences in tumor appearance. This study recruited patients with GBM from local hospitals with two MR vendors [31,32]. Regarding the texture features, GLRLM is a matrix of all the voxels within the same gray-level value. To determine texture matrix representations, the voxel intensity values within the volume of interest need to be discretized. Consequently, semiautomatic segmentation and specialized algorithms trained on limited datasets may not apply thoroughly to data acquired from different hospitals, MR acquisition protocols, and patient populations.

The third challenge is heterogeneous tumor morphology: diffuse astrocytomas often exhibit the essential neuropathological features of cellular pleomorphism, necrosis, and microvascular proliferation, but it is their irregular morphological profile that reduces the accuracy of gross tumor contouring. Notably, irregular rim enhancement surrounding the necrosis is the most complicated form for imaging annotation; by contrast, it would be advantageous when using semiautomatic segmentation to determine GBM shape in each patient. Moreover, GBMs are characterized by heterogeneous angiogenesis, cellular proliferation, cellular invasion, and apoptosis, which translate into diversifying grades of necrosis, solid enhancing tumor, peritumoral tissue, and peritumoral edema, which makes reliable imaging assessment challenging. Radiomic features incorporating machine learning techniques may, therefore, be well suited to solving such image-based problems of the accurate and expeditious interpretation of large-quantity and complex data that minimize semiautomated bias. These data can provide initial ground-truth estimates, which can then be refined by human experts for enhanced quality [33]. Machine learning is an efficient technique for analyzing radiomic features, as well as classifying the GBM tissue regions. Rathore et al. [34] used a radiomics signature of infiltration in peritumoral edema to predict subsequent recurrence in GBM. Beig et al. [35] proposed a radiomic risk score analysis which revealed that the “low-risk” and “high-risk” radiomic risk groups compared with tumor habitat (i.e., necrotic, enhancing tumor, edema) in their ability to predict progression-free survival on pretreatment MRI. Furthermore, we identified the features from the enhancing tumor and the tumor shape elongation of peritumoral edema region for high-risk groups, as previously shown in (Figure 2).

The small sample size is another limitation to this study; we have also retrieved 31 patients with GBM from TCIA database as our validation cohort to carry out a pilot study for more clinical subjects in rare cases (high aggressive brain tumors), and try to validate the performance on the whole dataset. Although our model achieved high accuracies in each region, this is merely due to the small sample size. However, we attempted to compare our results with real-world clinical experience and across different types of study for correlation with radiomics features’ accuracy and efficiency related to specific regions of GBM. Efficient radiomics-based classification of multi-parametric, to identify distinct tumor habitats MR images in glioblastoma, helps quantitative trait, texture analysis to develop features that provide novel clinical insights for personalized and precision medicine. It emerged as a powerful data-driven approach that can offer insights into clinically relevant questions related to diagnosis, prediction, prognosis, and the assessment of treatment response.

## 5. Conclusions

Radiomics classifiers integrating multiparametric MRI parameters may have potential in prognostication from the routine MRI exam, and may also be significantly associated with key biological processes that affect the response to chemotherapy in GBM, including the relative imaging signature of possible underlying tumor biological processes, to facilitate personalized precision medicine.

## Figures and Tables

**Figure 1 jcm-10-02030-f001:**
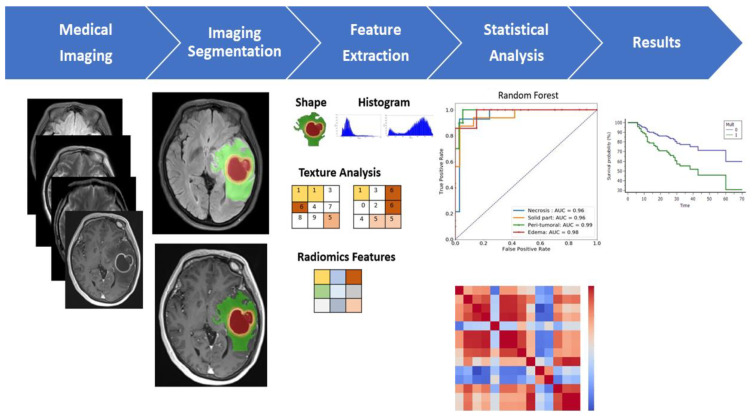
A multiparametric MRI-based radiomic analysis in step (1) medical imaging acquisition, (2) imaging segmentation, (3) feature extraction, (4) statistical analysis, and (5) results. The tumor ROI on all MR slices to extract the radiomic features. Features such as tumor shape, histogram, and texture features were extracted from the ROI to discriminate the biological processes of GBM habitats and facilitate personalized precision medicine. Note: GBM, glioblastoma multiforme; ROI, region of interest.

**Figure 2 jcm-10-02030-f002:**
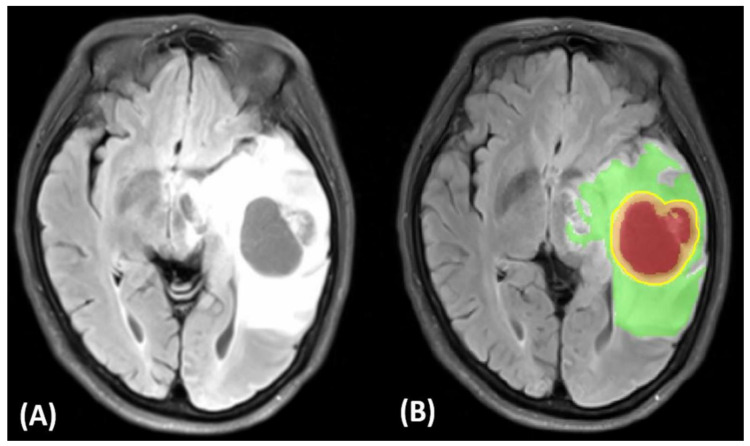
Tumor image signatures and habitats of a 61-year-old woman with left temporoparietal GBM. Illustration of ground truth and semantic features. (**A**) T2-FLAIR. (**B**) Red: necrosis, orange: solid part, yellow: peritumoral tissue, green: peritumoral edema.

**Figure 3 jcm-10-02030-f003:**
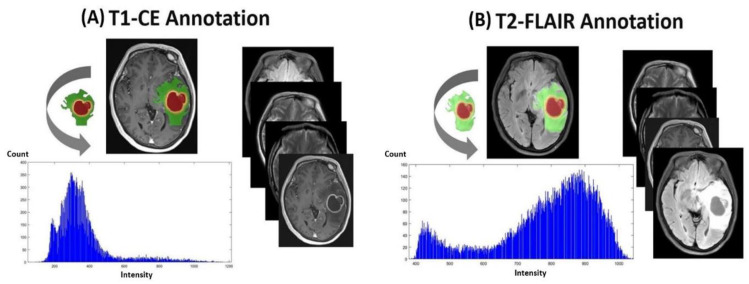
Illustration of ground truth, semantic features, and histogram on the raw MR images. (**A**) T1-CE annotation. (**B**) T2-FLAIR annotation.

**Figure 4 jcm-10-02030-f004:**
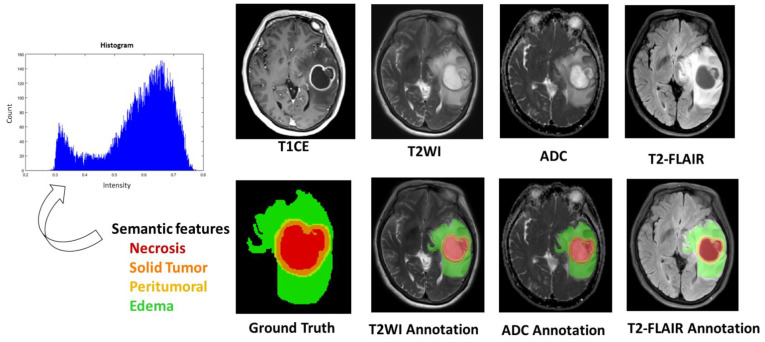
Radiomic signature shows representative T1-CE, T2-WI, ADC and T2-FLAIR images that demonstrate tumor habitats color-coded and overlaid, i.e., necrosis (**red**), solid part (**orange**), peritumoral tissue (**yellow**), and peritumoral edema (**green**). Each annotated area based on the raw MR images. Note: T1-CE, contrast-enhanced T1-weighted; T2-WI, T2-weighted images; T2-FLAIR, T2-weight fluid-attenuated in-version recovery; ADC, apparent diffusion coefficient images.

**Figure 5 jcm-10-02030-f005:**
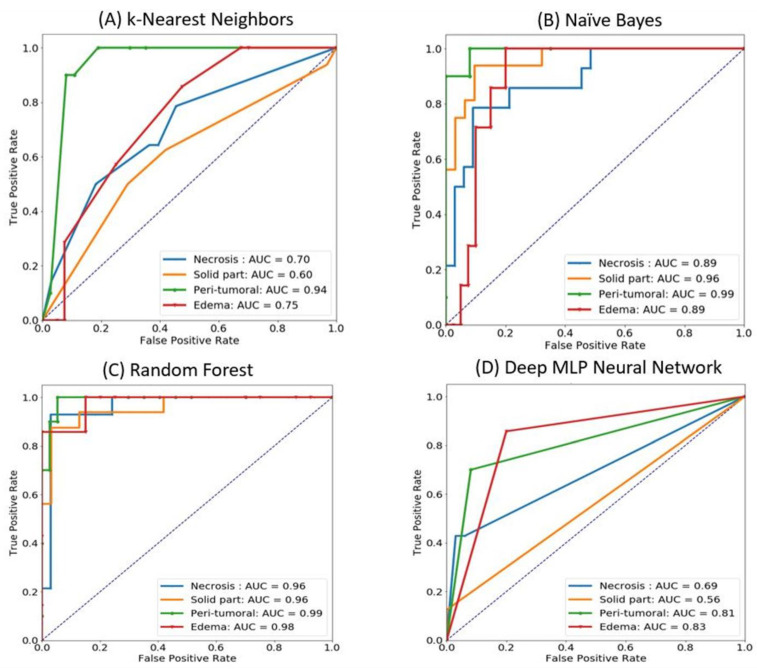
AUCs of classification of GBM regions using different machine learning/deep learning approaches: (**A**) k-Nearest Neighbors, (**B**) Naïve Bayes, (**C**) Random Forest, and (**D**) Deep MLP Neural Network.

**Figure 6 jcm-10-02030-f006:**
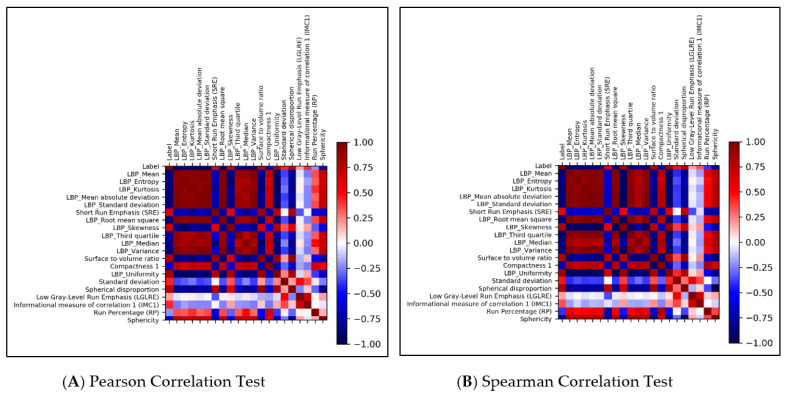
Correlation heatmap of 20 important radiomics features generated by random forest. (**A**) Pearson correlation test, (**B**) Spearman correlation test. The highly correlated features are LBP_Uniformity, surface to volume ratio, spherical disproportion, and LBP_Skewness.

**Table 1 jcm-10-02030-t001:** Patient demographics for two subsets of studies GBM from TCIA database as our validation cohort.

Cohort	Case Number	Gender	Mean Age (Years)	Required Image Contrasts	Model in Machine Learning
GBM Local Patients	23	13 Males, 10 Females	62.60 (Range: 42–83)	T1-CE, T2-WI, T2-FLAIR, ADC	Random Forest
GBM TCIA Database	31	16 Males, 15 Females	55.13 (Range: 18–84)	T1-CE, T2-WI, T2-FLAIR	Random Forest

**Table 2 jcm-10-02030-t002:** Performance results of classifying GBMs using the random forest algorithm.

ROIs	Recall	Precision	F1-Score	Accuracy
Necrosis	94.6	97.1	0.837	93.6
Solid part	93.6	94.3	0.745	90.4
Peritumoral	97.3	97.3	0.898	95.8
Edema	92.1	95.8	0.732	90.4

Random forest hyperparameters: *n*_trees = 500, *n*_features = 8.

## Data Availability

The data is a publicly available dataset that contains no linkage to patient identifiers. https://wiki.cancerimagingarchive.net/display/Public/TCGA-GBM (accessed on 29 March 2021).

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
