# Peer review of "A Multiparametric MRI-Based Radiomics Analysis to Efficiently Classify Tumor Subregions of Glioblastoma: A Pilot Study in Machine Learning"

_jcm, 2021, doi:10.3390/jcm10092030_

Round 1
Reviewer 1 Report
In the manuscript entitled "A Multiparametric MRI-based Radiomics Analysis to Effi-2ciently Classify Tumor Subregions of Glioblastoma: A Pilot 3Study in Machine Learning", the authors established a set of radionics features to efficiently classify patients with GBM and retrieved cerebral multiparametric MRI, including contrast-enhanced T1-weighted(T1-CE), T2-weighted, T2-weighted fluid-attenuated inversion recovery, and apparent diffusion coefficient images from local patients with GBM. The authors performed a leave-one-out cross-validation on the whole dataset, the different machine learning and deep learning techniques tested, random forest achieved the best performance. The authors concluded their research by stating that: "Multiparametric MRI-based radiomics model showed efficiently classify tumor subregions of GBMand suggests that prognostic radiomic features from routine MRI exam may also be significantly associated with key biological processes that affect response to chemotherapy in GBM".
The paper is very interesting. No significant criticalities emerged. Minor revisions will be required for the manuscript to be accepted.
Minors points:
Please improve the Abstract, in this way the objectives set and the added results are incomprehensible.
The organization of Figure 2 is not clear. Find a more optimal solution.
Please transform the data relating to the inclusion criteria used into a table, in this way they would help the reader in the interpretation.
Please create a workflow of the analysis procedures used and represent it graphically.
Please enter the values of pValue in table 3.
The choice to represent the Pearson and Spearman tests through heatmaps seems an interesting solution even if the goal of providing a complete solution has not been completely achieved. In this way, we can appreciate the data in a partial way. Furthermore, a sufficient description of the images is completely missing. The stair bar is too large and aesthetically annoying. You can improve the figure.
Author Response
Reviewer 1:
- The organization of Figure 2 is not clear. Find a more optimal solution.
Ans: Thank you! We changed High Resolution DPI OVER than 600. Hopefully that’s much better! Revised as requested.
- Please transform the data relating to the inclusion criteria used into a table, in this way they would help the reader in the interpretation.
Ans: Thank you! Revised as requested in Table 1.
|
Cohort |
Case Number |
Gender |
Mean Age (years) |
Required Image Contrasts |
Model in Machine Learning |
|
GBM Local Patients |
23 |
13 Males, 10 Females |
62.60 (Range: 42-83) |
T1-CE, T2-WI, T2-FLAIR, ADC |
Random Forest |
|
GBM TCIA Database |
31 |
16 Males, 15 Females |
55.13 (Range: 18-84) |
T1-CE, T2-WI, T2-FLAIR |
Random Forest |
Table 1. Patient demographics for two subsets of studies GBM from TCIA database as our validation cohort. Note: GBM, glioblastoma multiforme; TCIA, The Cancer Imaging Archive; T1-CE, contrast-enhanced T1-weighted; T2-WI, T2-weighted images; T2-FLAIR, T2-weight fluid-attenuated in-version recovery; ADC, apparent diffusion coefficient.
- Please create a workflow of the analysis procedures used and represent it graphically.
Ans: Thank you! Revised as requested in Figure 1.
- Please enter the values of pValue in table 3.
Ans: We only had table 1 in this manuscript. However, table 1 aimed to show the performance results among four different regions. We did not have any baseline model as well as did not have any comparison in this table. Therefore, we might not include the p-value in this table.
Thank you for comments!

Reviewer 2 Report
As the authors mentioned, accurate demarcation on magnetic resonance imaging (MRI) between the active tumor region and perifocal edematous extension is essential for planning stereotactic biopsy, GBM resection, and radiotherapy. In this manuscript, the authors used machine learning and deep learning techniques to link a set of selected multiparametric MRI-based radiomics features with the classification of GBM tumor subregions (necrotic core, solid part, peritumoral tissue, and peritumoral edema), thus provided a promsing data-driven approach for better understanding of GBM classification and novel clinical insights into GBM characteristics for personalized precision medicine.
Overall, the study is interesting and well presented. However, I have the following queries. The manuscript can be accepted after addressing the minor revisions noted.
1. The authors mentioned that total 1316 features were selected for each annotated area based on the raw MR images, including contrast-enhanced T1-weighted (T1-CE), T2-weighted (T2-WI), T2-weighted fluid-attenuated inversion recovery (T2-FLAIR), and apparent diffusion coefficient (ADC) images from local patients with GBM. Here, it would be better to supplement the representative T1-CE, T2-WI and ADC images that showing tumor habitats color-coded and overlaid. As well, please provide the illustration of ground truth, semantic features, and histogram on the representative raw MR images including T2-WI annotation and ADC annotation.
2. In terms of the sample used in this study (MRI images from 23 GBM patients), here wondering how many images were used here? As authors mentioned that some patients were undergone surgery, here wondering whether these images also included many that had been took after surgery? Or some from recurrent GBM? Please clarify this more clearly.
3. The authors brought up that “The features from the enhancing tumor and the tumor shape elongation of peritumoral edema region for high-risk groups from T1-CE.” in Abstract (Line 26-27). A little bit confused on this clarification, please explain this more clearly.
4. In Part 3.2, the authors mentioned that a set of 292 radiomics features were input to evaluate the performance of the GBM classification for the kNN, Naïve Bayes, random forest, and DNN models. Here please explain why just 292 features were used. And what the results would be if more features are used?
5. The same concern in table 1, which was noted that “n_trees = 500, n_features = 8”. First please explain this more clearly. As well, whether this result was just based on 8 features? If yes, why? And what these 8 features are?
6. In part 2.4, the equation for “Precision” (Line 167) is the repeated one and should be replaced with “F-1 score”.
Author Response
Reviewer 2:
- The authors mentioned that total 1316 features were selected for each annotated area based on the raw MR images, including contrast-enhanced T1-weighted (T1-CE), T2-weighted (T2-WI), T2-weighted fluid-attenuated inversion recovery (T2-FLAIR), and apparent diffusion coefficient (ADC) images from local patients with GBM. Here, it would be better to supplement the representative T1-CE, T2-WI and ADC images that showing tumor habitats color-coded and overlaid. As well, please provide the illustration of ground truth, semantic features, and histogram on the representative raw MR images including T2-WI annotation and ADC annotation.
Ans: Thank you! Revised as requested in Figure 4.
- In terms of the sample used in this study (MRI images from 23 GBM patients), here wondering how many images were used here? As authors mentioned that some patients were undergone surgery, here wondering whether these images also included many that had been took after surgery? Or some from recurrent GBM? Please clarify this more clearly.
Ans: (1) 276 raw data images for each patient it depends on patient brain size. T1CE: 192 Images, T2WI: 28 Images, T2-FLAIR: 28 Images, ADC: 28 Images. (2) some patients were undergone surgery and available preoperative MRI consisting of gadolinium-based T1-CE, T2-WI, T2-FLAIR, ADC images not included after surgery/recurrent GBM in this study.
- The authors brought up that “The features from the enhancing tumor and the tumor shape elongation of peritumoral edema region for high-risk groups from T1-CE.” in Abstract (Line 26-27). A little bit confused on this clarification, please explain this more clearly.
Ans: The shape features (i.e., sphericity, elongation, and convexity) of the peritumoral edema region and biological processes of cell proliferation, angiogenesis, and cell adhesion. Elongation of the edematous region (ratio between major and minor axes of the tumor) are indicative of tumor progression. We identified the features from the enhancing tumor and the tumor shape elongation of the peritumoral edema region for high-risk groups from T1-CE as shown in Figure 2 & 3.
- In Part 3.2, the authors mentioned that a set of 292 radiomics features were input to evaluate the performance of the GBM classification for the kNN, Naïve Bayes, random forest, and DNN models. Here please explain why just 292 features were used. And what the results would be if more features are used?
Ans: The reviewer is appreciated for valuable comments. We are sorry for this mistake, we aimed to say that we used 329 (Line 242) radiomics features for each region, thus we had totally 1316 features for all four regions. We have already revised this part in the revised manuscript.
- The same concern in table 1, which was noted that “n_trees = 500, n_features = 8”. First please explain this more clearly. As well, whether this result was just based on 8 features? If yes, why? And what these 8 features are?
Ans: As mentioned in the paper, we performed hyperparameter tuning to find the optimal set of features, and it is noticed that “n_trees=500, n_features=8” is the best suit for Random Forest. We also compared the predictive performance among different machine learning models and Random Forest outperformed the other ones. Therefore, we reported the Table 2 using Random Forest classifier. Also, please be noticed that “n_features” here is a hyperparamter of Random Forest algorithm, it is not stand for our radiomics features using in this study.
- In part 2.4, the equation for “Precision” (Line 167) is the repeated one and should be replaced with “F-1 score”.
Ans: YES!! That should be replaced with “F-1 score” (Line 167). Thank you for pointing out the mistakes.
Thank you for comments!